# RNA-Binding Proteins Hold Key Roles in Function, Dysfunction, and Disease

**DOI:** 10.3390/biology10050366

**Published:** 2021-04-24

**Authors:** Sophia Kelaini, Celine Chan, Victoria A Cornelius, Andriana Margariti

**Affiliations:** Wellcome-Wolfson Institute for Experimental Medicine, School of Medicine, Dentistry and Biomedical Sciences, Queen’s University Belfast, 97 Lisburn Road, Belfast BT9 7BL, UK; s.kelaini@qub.ac.uk (S.K.); cchan10@qub.ac.uk (C.C.); vcornelius01@qub.ac.uk (V.A.C.)

**Keywords:** RNA binding protein, splicing factor, translation regulator, disease, stress granules

## Abstract

**Simple Summary:**

Chronic hyperglycemia manifests in a variety of different micro- and macrovascular disorders such as diabetes mellitus and cardiovascular disease and has been shown to have links to post-transcriptional dysregulation. Equally, the development and progress of other devastating disorders such as tumorigenesis and neurodegenerative disease have also been associated with dysfunction of molecules involved in epigenetics such as RNA-binding proteins. Recent advances, especially on an analytical systemic level, have revealed new roles for these proteins and their contribution in maintaining the balance between normal function and dysfunction/disease.

**Abstract:**

RNA-binding proteins (RBPs) are multi-faceted proteins in the regulation of RNA or its RNA splicing, localisation, stability, and translation. Amassing proof from many recent and dedicated studies reinforces the perception of RBPs exerting control through differing expression levels, cellular localization and post-transcriptional alterations. However, since the regulation of RBPs is reliant on the micro-environment and events like stress response and metabolism, their binding affinities and the resulting RNA-RBP networks may be affected. Therefore, any misregulation and disruption in the features of RNA and its related homeostasis can lead to a number of diseases that include diabetes, cardiovascular disease, and other disorders such as cancer and neurodegenerative diseases. As such, correct regulation of RNA and RBPs is crucial to good health as the effect RBPs exert through loss of function can cause pathogenesis. In this review, we will discuss the significance of RBPs and their typical function and how this can be disrupted in disease.

## 1. Introduction

RNA-binding proteins (RBPs) are critical RNA regulators responsible for modulating post-transcriptional events in the cell. RBPs can recognize and interact with binding motifs called RNA recognition motifs (RRM) and/or RNA structure to form ribonucleoprotein (RNP) complexes for the regulation of various RNA processes such as RNA stability, alternative pre-mRNA splicing, mRNA decay, translocation, post-translational nucleotide modifications, and RNA localization (Figure 1) [1,2].

The mRNA life cycle from newly transcribed mRNA molecules to the generation of functioning mature mRNA transcripts is an intricate system which is governed by many different RBPs. In the human genome there are at least 1200 verified RBPs as well as several newly discovered ones [3]. RBP-RNA binding occurs at the RNA binding domain (RBD) which are found within the coding sequences (intron and exon domains), 5′ untranslated regions (5′UTR) and 3′ untranslated regions (3′UTR) of RNA. RBP binding produces various effects on RNA splicing, transcription efficiency, stabilization, and more. For example, RBPs can interact with target binding sites within the coding region to facilitate alternative splicing while RBP-RNA interactions within the 3′UTR domain can inhibit or induce mRNA decay, as well as mediate RNA stabilization. Conversely, lack of RBP binding to 3′UTR targets can destabilize mRNA molecules [4]. In addition, RBPs have key regulatory roles in important RNA maturation events such as polyadenylation, the addition of the 5′ cap, and pre-mRNA alternative splicing, all of which are vital for the expression of functioning, mature RNA. However, RBPs can initiate degradation and RNA decay through deadenylating enzymes to remove the 3′ poly-A tail, and decapping enzymes to remove the 5′cap, therefore enabling RNA degradation [5]. Several RBPs have been implicated in human disease, from vascular conditions such as diabetes and heart disease to cancer and neurogenerative disorders. Here, we showcase selected examples of RBP dysregulation and their subsequent contribution to the development of human diseases (Table 1).

## 2. RBPs in the Pathogenesis of Diabetes and Cardiovascular Disease

Diabetes mellitus (DM) is an increasingly prevalent global health burden [6]. DM is a lifelong disease, characterized by chronic hyperglycemia. DM is highly associated with an increased risk of debilitating secondary morbidities manifesting in macrovascular disease (atherosclerosis, ischemic stroke, coronary artery disease) and microvascular disease (diabetic retinopathy, neuropathy, and nephropathy) [7]. Close to 10% of worldwide diabetes diagnoses are categorized as Type 1. The remaining majority are diagnosed as Type 2 [8], where cells become increasingly resistant to insulin action [9], leading to impaired glucose homeostasis with cells unable to internalize circulating blood glucose. Chronic hyperglycemia causes systemic damage to the vasculature triggering multisystemic conditions such as cardiovascular disease (CVD). Additionally, due to the frequency of CVD occurrence in diabetes, it is often considered a CVD in itself. Currently, there is no curative therapy available for diabetes-associated CVD. With rising rates of diabetes, there lies a deepening need for knowledge into the mechanisms behind hyperglycemia-related cardiovascular damage. In vascular endothelial cells (ECs), hyperglycemia has been determined to contribute to a substantial change of gene expression. Transcriptomic analytical assays have uncovered a wide variety of candidate genes implicated in cellular functions such as angiogenesis, coagulation, vascular tone, adhesion, and more. This vascular EC gene expression is tightly controlled by transcriptional and post-transcriptional regulatory mechanisms, the latter including regulation of pre-mRNA to mRNA processing, transport, decay and protein translation [10]. Precise regulation of these complex post-transcriptional modifications in the RNA network is crucial for the normal function of vascular ECs and the endothelial system. In diabetes, a plethora of RBP-regulated RNA networks are involved in the dysfunction of the vascular endothelium [11]. In this section, we will review some of the most common RBPs dysregulated in the pathogenenesis of DM and CVD and their epigenetic effects.

For instance, RNA Binding Fox-1 Homolog 2 (RBFOX2), regulates alternative splicing and is upregulated in the diabetic heart, controlling splicing of genes involved in diabetic cardiomyopathy by binding to target RNA motifs associated with protein trafficking and cell apoptosis [12,13]. Additionally, Human Antigen R (HuR) also known with its alternative name ELAVL1, is a ubiquitously expressed RBP, which is upregulated and activated under high glucose and in diabetes [14]. HuR binds to specific domains known as AU-rich elements (ARE) in the 3′UTRs of target genes that play a role in inflammation and diabetic nephropathy [15,16,17]. Once it is activated, it translocates to the cytoplasm to bind its mRNA targets affecting their stability and translation [18]. Tristetraprolin (TTP) binds to 3′ UTR ARE region that results in mRNA destabilization and decay [19]. TTP is minimally expressed in healthy aortas but significantly heightened in affected macrophage foam cells as well as ECs of atherosclerotic lesions [20]. In another example, Quaking (QKI), an RBP member of the Signal Transduction and Activation of RNA (STAR) protein family and some of its isoforms—namely QKI5, QKI6, QKI7—have been associated with vascular development [19]. In our lab, we have previously shown that, compared to controls, there are reduced QKI5 levels in cardiac vessels of diabetic mice, therefore displaying the key status of QKI5 within the diabetic framework of vessel dysfunction. In addition, as we have also reported [21], QKI5 played a crucial role in differentiating ECs from induced pluripotent stem cells (iPSCs) via stabilization of VE-cadherin and Vascular Endothelial Growth Factor Receptor 2 (VEGFR2) activation through Signal Transducer and Activator of Transcription 3 (STAT3) signaling. Furthermore, we showed QKI-7 to bind and promote mRNA target degradation such as of VE-cadherin, while the knockdown of QKI7 in a diabetic mouse model of hindlimb ischemia significantly restored reperfusion and blood flow in vivo [22].

RBPs are also implicated in the dysfunction of ECs under diabetic conditions in relation to their association with non-protein coding RNAs (ncRNAs), which include long noncoding RNAs (lncRNAs). The latter are, in fact, responsible for the preponderance of gene transcripts and act as positive or negative regulators based on their interactions with RBPs [23]. The importance of the RBP and lncRNAs system as a fundamental part of healthy cellular function through regulation of epigenetic machineries is largely acknowledged [24]. In recent years, data from numerous studies has showed a correlation between abnormal levels of lncRNAs and different diseases such as diabetes. For example, the levels of Metastasis-Associated Lung Adenocarcinoma Transcript 1 (MALAT1) were significantly elevated both in vivo (in retinal ECs of a streptozotocin (STZ) diabetic rat model) and in vitro when human umbilical vein endothelial cells (HUVECs) were treated with high glucose. Short hairpin RNA (shRNA) knockdown of MALAT1 reduced vascular dysfunction and also decreased reactive oxygen species (ROS) levels in hyperglycemic ECs signifying its connection with diabetic retinopathy and EC dysfunction [25,26]. Similarly, under diabetic conditions, myocardial infarction associated transcript (MIAT) lncRNA is elevated, as data from studies of diabetic retinas and high-glucose treated ECs have shown. Furthermore, knockdown of MIAT reversed the dysfunction [27]. More such examples of lncRNA dysregulation in diabetic conditions exist as in the case of increased antisense noncoding RNA in the INK4 Locus (ANRIL) [21] promoting pathogenic angiogenesis [28] or in the case of Maternally Expressed Gene 3 (MEG3), which had reduced levels in the retinal ECs as shown in an STZ model of diabetic mice [29].

The above examples demonstrate the close association of lncRNAs and RBPs and the importance of this relationship in diabetic pathogenesis. At the same time, it is becoming clearer that advances in RNA biology uncover more roles for RBPs in the progression of diabetes where the dysregulation of RBPs controlling alternative splicing, RNA decay, and destabilization can have disastrous downstream effects on cardiovascular genes.

## 3. RBPs and Their Role in Cancer Development and Progression

RBPs are tightly associated with tumorigenesis [30] and dysregulated RBPs have been reported by many studies to play a critical role in cancer [31]. For example, mutations in the splicing factor U2 Small Nuclear RNA Auxiliary Factor 1 (U2AF1), affects pre-mRNA splicing and contributes to the progression of cancer such as myelodysplasia (MDS) [32]; even with a single mutation such as S34F [33], which affects hundreds of mRNAs. In another example, Splicing Factor 3b Subunit 1 (*SF3B1)* has been found to be mutated in a lot of patients suffering from chronic lymphocytic leukemia (CLL) [34,35]. Mutated *SF3B1*, along with mutated *U2AF1*, Serine and Arginine Rich Splicing Factor 2 (*SRSF2)* and Zinc Finger CCCH-Type, RNA Binding Motif And Serine/Arginine Rich 2 (*ZRSR2)* genes, is also very frequently observed in MDS [36] and has been linked to poor survival rate [34]. Negative Elongation Factor E (NELFE) was also reported to promote hepatocellular carcinoma (HCC) progression by augmenting MYC signaling and selectively controlling MYC-associated genes [37].

Despite the extensive studies on a selection of individual RBPs, the role of RBPs in cancer continue to appear obscure and more studies are still needed to define in a clearer way the landscape of RBP expression in human cancers. Consequently, recent studies are now focusing on integrating the expression profiles of multiple RBPs to universally and analytically examine their mechanism of action in many different cancers [33]. Studies focusing on building a thorough expression profile of several hundreds of RBPs in 16 different types of human cancer uncovered that RBPs are primarily upregulated in cancers compared to downregulated ones and that their dysregulation can be influenced by the tumor microenvironment. Specifically, the number of RBPs found to be consistently upregulated in cancer were 109 compared to downregulated RBPs which were just 41 [38]. As it is widely known, an mRNA’s destiny is primarily defined by its interactions with RBPs. In cancer, dysregulation in the RBP-mediated RNA stability, localization, and post-transcriptional events can have an effect in cancer profiles [39].

Variations in an RBP’s expression or localization has the capacity to impact oncogene expression levels or those of tumor-suppressor genes. They can also influence genes important to genome stability. As a result, different transcriptomic and cellular phenotypes arise under the influence of RBP-centered gene regulation, such as differences in proliferation or apoptosis, as well as in other functions like angiogenesis or epithelial to mesenchymal transition (EMT). Eventually all these can, in turn, give rise to different profiles of cancer invasion and metastasis as well as different cancer prognoses. It is therefore becoming more and more apparent that RBPs can act as prospective targets for future cancer treatments [40].

One of the key roles of RBPs lies in their regulatory involvement in micro-RNA (miRNA) biogenesis and subsequent maturation. The close relationship between dysregulation of miRNA processing and RBP expression can lead to mRNAs alterations that can contribute to cancer [41]. For example, the RBP LIN28 and its related *Lin28*/*let-7* pathway can lead to cancer development and progression by promoting increased cell proliferation, invasion, or angiogenesis [42]. The cellular areas where mRNA or noncoding RNA such as lncRNA are localized can also alter protein expression; cancer associated RBPs bind to these RNA targets to coordinate to guide localization and translation [43,44]. For instance, the RBP Cytoplasmic Polyadenylation Element Binding Protein 1 (CPEB1) controls Zonula occludens-1 (*ZO-1)* mRNA localization. When there are diminished CPEB1 levels, this decrease causes *ZO-1* mRNA to scatter within the cell causing epithelial cell polarity impairment [45], which is linked to metastatic risk [46]. RBPs can also exert effect on mRNA stability. The latter relies on its 5′ -terminal 7-methylguanosine cap as well as its 3′ poly(A) tail, both of which protect mRNAs from decay and also stimulate translation [47]. mRNA decay can transpire through decapping or deadenylation of the poly(A) tail) of the mRNA targets which are transported to cytoplasmic aggregations such as stress granules [48]. In cancer, RBPs, including HuR, TTP, and Insulin Like Growth Factor 2 MRNA Binding Protein 1 (IGF2BP), can affect the stability of target RNAs [49].

Another characteristic of RBP dysregulation in cancer is gene regulation and translational control. RBPs, by binding to the 5′- or 3′-UTR of RNA, are implicated in translation in a varying binding capacity which affects translation efficiency [50]. For example, Eukaryotic Translation Initiation Factor 4E (eIF4E), which has increased levels in cancer, promotes tumorigenesis through mRNA-mediated cellular functions that include, for example, proliferation or angiogenesis [51]. Alternative splicing also constitutes a vital process through which dysregulated RBPs may exert an effect in cancer development through different mRNA splice variants resulting in protein diversity and aberrant splicing forms in cancer [52]. Such modifications can enhance the action of oncogenes or, conversely, quell tumor-suppressive genes [53,54]. For example, SRSF1 is known to modify the splicing of the protooncogene Recepteur d’origine nantais (Ron) as well as the tumor suppressor gene Bridging Integrator 1 (*BIN1)* [54,55]. Lastly, alternative polyadenylation (APA) of target mRNAs, a process involving the modification of 3′UTR length, can be seen in cancer-related genes (oncogenes and tumor suppressors), promoting cancer development by altering their expression through RBP mediated regulation [56]. RBPs specifically regulate APA by enlisting or competing with other proteins within the polyadenylation machinery [57]. CPEB1, for example, alters the 3′-UTR length of target mRNAs and, in turn, modifies the associated gene expression signatures. As a result, decreased CPEB1 expression lengthens the poly(A) tail resulting in increased Matrix Metallopeptidase 9 (MMP9) mRNA translation in breast cancer [46]. 

## 4. RBPs and Their Role in Neurodegenerative Disease

Even though the functional mechanisms of RBPs are still not fully elucidated, more recent evidence has indicated that RBPs are key players in the preservation and integrity of neurons. Any defects and alterations in the function of RBPs and in RNA metabolism arising from mutations can cause several neurodegenerative diseases that affect the central nervous system, such as frontotemporal lobar degeneration (FTD), amyotrophic lateral sclerosis (ALS), fragile X syndrome (FXS), or spinal muscular atrophy (SMA). Other diseases usually associated with aging and that can be affected by RBP dysregulation include Alzheimer’s disease (AD) and Parkinson’s disease (PD). The increasing aging global population has additionally resulted in an increase in the number of worldwide dementia cases, despite a relative decrease in developed countries [58]. In the case of ALS for instance, analytical investigations have revealed a strong genetic relationship between mutations of RBP-encoding genes like Ataxin 2 (*ATXN2)*, Heterogeneous Nuclear Ribonucleoprotein A1 (*hnRNPA1*), Matrin 3 (*MATR3*) or TIA1 Cytotoxic Granule Associated RNA Binding Protein (*TIA-1*), and development and progression of the disease [59,60]. Likewise, in FTD, which shares many common characteristics with ALS [61,62], fragmentation of RBPs such as TAR DNA-binding protein 43 (TDP-43) in the cytoplasm, has been shown to advance the onset of the disease [63,64]. In the case of SMA, a serious motor neuron disease, small molecule drug analogs of RG-7916 (SMN-C2 or -C3) were found to selectively regulate alternative splicing of Survival of Motor Neuron 2 (*SMN2*) by binding to the gene’s pre-mRNA and increasing the affinity of the RBP Far Upstream Element Binding Protein 1 (FUBP1) to it [65]. Another neurological syndrome Paraneoplastic opsoclonus-myoclonus ataxia (POMA) is caused by autoantibody secretion against the RBP neuro-oncological ventral antigen 1 and 2 (Nova 1, Nova 2) [66], which are neuron-specific found in the nucleus and regulate RNA splicing [67]. In another example, myotonic dystrophy (MD) which commonly presents in patients as muscular degeneration, is also characterized by aberrant RNA splicing; CUG triplet repeat (CUGBP) has been specifically linked to MD through its interaction with myotonic dystrophy protein kinase (DMPK) mRNA [68].

These types of neurological diseases usually present with aggressive and irreversible characteristics that can prove devastating, and on many occasions even fatal, such as permanent neuron loss, which involves neural cells such as microglia and astrocytes. In neurodegenerative disease a great deal of attention has been concentrated on the different protein aggregates; however, it is of utmost importance to also focus on additional avenues that involve RNA and post transcriptional modifications as a pathogenic component of neurodegenerative disease [69,70]. In neurons, looking at the high incidence of RNA transport granules may explain why RBP dysfunction can initiate neuronal disease. The RNA granules creation and aggregate formation in a cell’s cytoplasm has been considered to be pathogenic in nature. During cellular stress, RBPs like those with low complexity domains (LCD), such as FUS RNA binding protein (*FUS*) or *hnRNPA1* translocate from the nucleus, where they are usually present, to the cytoplasm and localize in granules [71,72]. Once there, they transiently form droplet organelles [73] with different functions based on their components [74]. Higher RBP concentrations can change these functions and lead to the polymerization of LCDs and the creation of amyloid-like fibers and insoluble aggregates [75]. Studies on these aggregates are giving rise to hypotheses that in neurons affected by dementias such as ALS or AD, disturbances such as mutations in RBPs play a role in impairing their regular physiological function. In the case of dementias such as AD, a study on RBP-containing stress granules showed their elevated accumulation in the brains of transgenic mice used as a model of tauopathy [76]. Furthermore, these granules have an interconnecting role with miRNAs, since the latter interact with RBP to regulate protein translation [77], adding an additional layer of complexity.

In general, mutations in the proteins associated with disease increase their propensity for higher aggregation, shifting the balance towards increased creation of more stable, less soluble and, thus, more persistent, stress granules, including secondary granules, which are usually associated with disease. Equally, approaches in neuroprotective therapeutics are directing their efforts against pathogenesis by reducing the creation of stress granules and restoring the balance [78].

## 5. RBP-Based Therapeutics and Future Directions

It is widely known that RBPs are crucial players in epigenetic post-transcriptional gene regulation. A plethora of studies has revealed the interconnection between RBPs and mRNA including a complex network of fifty thousand interactions [3]. In Figure 2, a summary of RBP targeting therapeutic strategies is presented. Since these RBP-mediated RNA networks can drive vascular pathogenesis, steering therapeutic investigations towards the discovery of putative candidate RBPs is the way forward towards treating vascular abnormalities and endothelial dysfunction prevalent in diabetes and heart disease. It is noteworthy that, even individually, RBPs can be utilized as potential targets for prospective treatments. As such, the RBP TTP is protective in inflammatory conditions involving diabetes and atherosclerosis by acting as a mediator for pro-inflammatory cytokine degradation [79]. For example, in the case of increased expression of tumor necrosis factor (TNF), a hallmark indicator of chronic inflammation, TNF homeostasis is regulated by TTP and, more specifically, by a post transcriptional regulatory positive or negative feedback loop [80]. An additional illustration of a prospective treatment strategy on the subject of heart disease is Poly(C)-binding protein 2 (PCBP2) which was shown to be reduced in the diseased heart in humans as well as in hypertrophied hearts in mice. Moreover, its silencing in neonatal cardiomyocytes, in particular, supported angiotensin II (Ang II)-induced hypertrophy, whilst the reverse effect was achieved through its overexpression [81,82].

Regarding cancer therapeutics, the role of RBP dysfunction in its initiation and spread is well known. What is not fully understood, however, is how to wholly and safely utilize the available RBP based approaches. Previously, it was not thought possible to target RBPs in cancer due to not being able to target them directly with specific drugs. Lately, however, it is becoming increasingly clearer that it might be feasible to target RBPs indirectly and with varied approaches. These approaches include the use of small molecules, which is the most common RBP-targeting tactic. Small molecules can hinder RBP-RNA interaction by, for example, binding to RBDs such as in the case of small molecules targeting eIF4E [83]. Another anti-cancer strategy involves the use of an oligonucleotide-based strategy which includes short antisense oligonucleotides (ASOs), a few of which have previously been approved for the treatment of other diseases such as hyperlipidemias or viral infections [84]. Currently, no such drugs to our knowledge are approved for the treatment of cancer, however, the potential of many are presently being evaluated in pre-clinical and clinical trials. For example, ISIS 183750 targeting eIF4E has been tested in clinical trials in patients with advanced stage colorectal cancer [85]. Furthermore, therapies involving siRNA have been employed for the targeting of RBPs such as HuR, a great a potential target in ovarian cancer [86]. Other strategies used as anti-cancer agents include aptamers with a mechanism similar to antibodies. Consequently, a clinical study found that addition of AS1411 to chemotherapy regimen in AML patients showed improved response [87]. Additionally, various synthetic peptides that target and bind RBPs such as elF4E have been developed and shown to exert strong antitumor effects in mouse models of ovarian cancer [83]. Moreover, circular RNAs (circRNAs) have been shown to act as RBP ‘sponges’ such as (Poly(A) Binding Protein Nuclear 1 (PABPN1)-derived circRNA binding to HuR [88,89]. Finally, whilst the Clustered Regularly Interspaced Short Palindromic Repeats/CRISPR Associated Protein 9 (CRISPR/Cas9) system has not been used in cancer treatment, yet it similarly holds great promise as a future anti-cancer approach [90].

In current neurodegenerative treatment strategies, typical therapeutic approaches are usually focused on limiting the generation of toxic aggregates in the cytoplasm. These approaches include activation of the heat shock response (HSP), utilization of HSP104 disaggregases or regulation of autophagy. For example, HSP104 disaggregase corrected many [PSI+] prions which are based on amyloid formation [91] while its modified version improved the mode of action compared to wild-type, efficiently suppressing TDP-43 or FUS based toxicity in yeast cells [92]. In rat neurons overexpressing TDP-43, another HSP, the Heat Shock Factor 1 (HSF1), prevented cytoplasmic aggregation of TDP-43 and also reduced toxicity in TDP-43 overexpressing human bone marrow neuroblasts [93]. Likewise, data from studies on autophagy activators showcased that they have the capability to rescue motor dysfunction in a transgenic mouse model of FTD. In a similar fashion, autophagy induction boosted TDP-43 turnover and enhanced the survival of neuronal cells in models of ALS [94,95]. These approaches offer a prospective therapeutic strategy through the elimination of toxic cytoplasmic aggregates. However, until the full elucidation of the mechanisms that trigger RBP based neurogenerative disease have been uncovered, it is challenging to define with absolute certainty the best therapeutic interventions delivering the biggest therapeutic effect in patients.

## 6. Conclusions

In conclusion, disruption to the function of RBPs and the ensuing post-transcriptional dysregulation in gene expression can lead to significant events in the development and progression of distinct human diseases. Any associated therapeutic approaches are still limited by the ambiguity surrounding the roles of RBPs which have not been fully understood yet and are still under investigation, given the complexity of their interaction with other cellular networks, pathways, and processes involved in disease. Nevertheless, newly emerging technologies that involve high throughput analyses allow for the uncovering and deciphering of further interconnections and the discovery of new RBP and RNA targets. The results of such analysis will shed light on the mechanisms underpinning human disorders that currently plague patients worldwide and provide assurance on the efficacy and safety of any novel corrective options. Future medicinal advances will grace us with the possibility of using these remedial strategies in a clinical setting.

## Figures and Tables

**Figure 1 biology-10-00366-f001:**
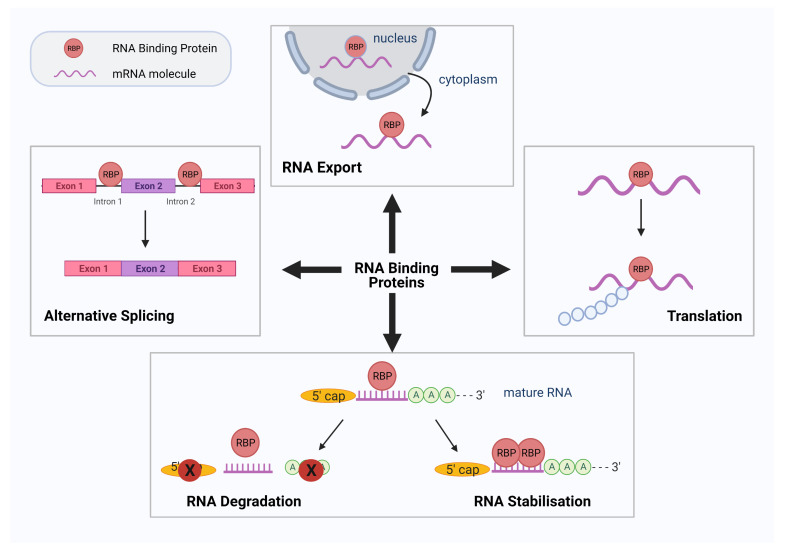
Schematic diagram summarizing the various roles of RNA binding proteins (RBPs). RBPs have numerous roles in RNA processing and translation. Four such functions of RBPs are demonstrated diagrammatically above: alternative splicing, RNA export, protein translation, RNA degradation, and stabilization. Figure created using Biorender.com.

**Figure 2 biology-10-00366-f002:**
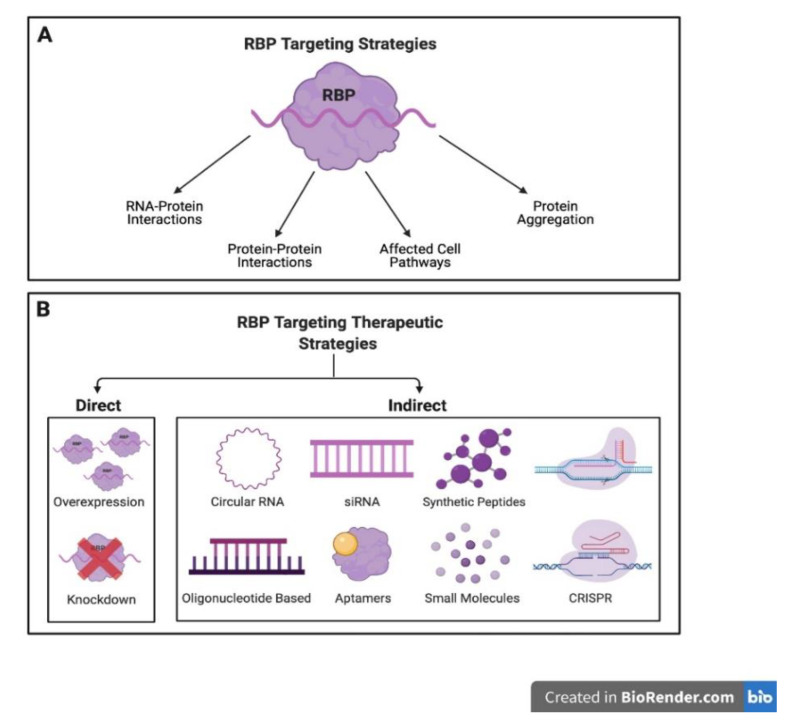
(**A**) Schematic diagram displaying RBP targeting strategies that may involve RNA-protein or protein-protein interactions, protein aggregation and cell pathways. (**B**) Current RBP-based targeting therapeutic strategies focus on either the manipulation of a specific RBP or an RBP-RNA interaction and so can be categorised as either direct or indirect approaches, respectively. Direct therapeutic strategies revolve around the knockdown or overexpression of a particular RBP. Indirect approaches, on the other hand, including the use of circular RNA, siRNA, synthetic peptides, oligonucleotide based, aptamers, small molecules, and CRISPR, can be designed to either inhibit the interaction of an RBP with RNA by inducing degradation, to suppress enzymatic activity, to block post transcriptional modifications or through binding to outcompete a chosen RBP. Figure created using Biorender.com.

**Table 1 biology-10-00366-t001:** Table displaying disorders arising from RNA-binding protein (RBP) dysfunction. Dysregulation of RBPs can lead to diseases such as cardiovascular and peripheral vascular disease, diabetes, cancer, and neurodegenerative disease.

RNA Binding Protein		Functions in Pathology	Disease Outcomes
RNA Binding Fox-1 Homolog 2	RBFOX2	Regulation of alternative splicing	Diabetic cardiomyopathy via alternative splicing defects of genes important for healthy cardiac regulation
Human Antigen R/ELAV Like RNA Binding Protein 1	HuR/ELAV1	Inducement of RNA stabilization and promotion of mRNA translation via binding to 3′UTR AREs	Diabetic nephropathy via binding of target genes such as SNAIL and FOS which contribute to EMT and nephropathy in diabetic conditions
Tristetraprolin	TTP	Inducement of RNA destabilisation and decay via binding to 3′UTR AREs	Atherosclerosis progression, and inflammation in TTP-deficient ECs
Quaking	QKI	Enablement of mRNA degradation	Diabetic EC dysfunction via degradation of targets such as VE-cadherin
U2 Small Nuclear RNA Auxiliary Factor 1	U2AF1	Mutations associated with disruption to pre-mRNA alternative splicing	Cancer progression via differential splicing of cancer-relevant gene targets in MDS
Mutated Splicing Factor 3b Subunit 1	SF3B1	Mutations associated with disruption to pre-mRNA alternative splicing	Cancer progression in CLL
Negative Elongation Factor E	NELFE	Inducement of mRNA stabilisation of protooncogenes	Cancer progression by stabilization of MYC-associated genes and MYC signalling in HCC
Lin-28 Homolog A	LIN28	Blocking of miRNA processing and maturation	Cancer development and progression via promotion of several cellular functions involved in cell proliferation, invasion, and angiogenesis
Cytoplasmic Polyadenylation Element Binding Protein 1	CPEB1	Enablement of mRNA localization Inducement of alternative polyadenylation	Cancer progression via promotion of cancer cell migration CPEB1 deficiency associated with cancer development
Insulin Like Growth Factor 2 MRNA Binding Protein 1	IGF2BP	Inducement of mRNA stability, translocation, and translation	Cancer progression via stabilization and translation of cancer-relevant mRNA
Eukaryotic Translation Initiation Factor 4E	eIF4E	Regulation of mRNA translation	Promotion of tumorigenesis by translation of protooncogenes, and malignancy-related factors
Serine/Arginine-Rich Splicing Factor 1	SRSF1	Regulation of alternative splicing	Cancer progression via splicing of protooncogenes and tumor suppressor genes
Ataxin 2	ATXN2	Mutations in genes elevated in neurodegenerative disorder	Progression and development of neurodegenerative disorder ALS
Heterogenous Nuclear Ribonucleoprotein A1	hnRNPA1		
Matrin 3	MATR3		
TIA1 Cytotoxic Granule Associated RNA Binding Protein	TIA-1		
TAR DNA-binding protein 43	TDP-43	Fragmentation and formation of inclusion bodies	Promotion of neurodegenerative disease advancement in ALS
FUS RNA Binding Protein	FUS	Regulation of RNA translocation, and localization in stress granules	Neuronal disease onset by stress granule aggregation
Neuro-oncological ventral antigen 1 and 2	Nova 1 and 2	Regulation of alternative splicing	POMA onset by autoantibody secretion
Far Upstream Element Binding Protein 1	FUBP1	Regulation of alternative splicing	Involvement in SMA by increasing FUBP1 affinity to SNF1 pre-mRNA.

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
