# Peer review of "RNA-Binding Proteins Hold Key Roles in Function, Dysfunction, and Disease"

_biology, 2021, doi:10.3390/biology10050366_

Round 1

Reviewer 1 Report

This is an excellent review of the literature concerning the role of RBPs in health and disease. The review is well-written and concise. Furthermore, the review explains with great detail the cellular mechanisms involving RBPs. Below, I provide some comments that I hope will improve this otherwise nice review. 

Comments:

The first three paragraphs of section 2 can be summarized into a short one because this review is not about DM and CVD.

In section 2, the authors provide a long list of RBPs linked to DM and CVD, but it is unclear why the authors chose to focus on them. 

Figure 2 is unnecessary because it does not provide an in-depth overview of the role of RBPs in the pathological conditions that the authors are describing. Instead of Figure 2, authors could provide a summary table to list the pathological conditions and the corresponding RBPs that are related to them.  

Section 4 is not exhaustive. No mention of SMA.

Reviewer 2 Report

In the current review article entitled “RNA-Binding Proteins Hold Key Roles in Function, Dysfunction, and Disease” by Kelaini et al the authors discussed the function of RBP in several disease conditions. This review will be a useful reference for readers in the field and researchers in common. The review is written very well; however, I have a few concerns and suggestions as mentioned below.

  • As a whole, the review is broken down into several small paragraphs which breaks the fluency in reading. It will be better if the authors can rearrange the text to address this.
  • The review is interesting and includes the name of several proteins, it will be easy to refer to if the authors include a table representing the RBP participating in diseases discussed in this review
  • Authors can include a figure on therapeutic strategy targeting RBP if possible

The text part of eth manuscript needs to check for spelling and typos font and italics, some of the minor typos are mentioned below

Minor suggestions

  • Abstract: line 7, number of diseases that include
  • Line 8 neurodegenerative diseases 
  • Page 3 section 2 title: RBP not RBB
  • Figure 2 legend – Protein – typo
  • Section 3: SF3B1 expansion (Splicing Factor 3b Subunit 1) should appear when it is mentioned first. U2AF is represented in a different format in different places
  • Section 3 paragraph 3, authors generalize in upregulation of RBP in cancer studies in line 2. Whereas in line 4 they also mentioned specifically RBS is are both up and downregulated, this contradicts while reading. It would be better to alter the second line accordingly
  • Page 6 ZO 1 expansion should appear when it is mentioned first.
  • Section 3: line 2 font/size error in "cancer"
